# OPENT2M: NO-FRILL MOTION GENERATION WITH OPEN-SOURCE, LARGE-SCALE, HIGH-QUALITY DATA

## ABSTRACT

Text-to-motion (T2M) generation aims to create realistic human movements from text descriptions, with promising applications in animation and robotics. Despite recent progress, current T2M models perform poorly on unseen text descriptions due to the small scale and limited diversity of existing motion datasets. To address this problem, we introduce **OpenT2M**, a **million-level**, **high-quality**, and **open-source** motion dataset containing over 2800 hours of human motion. Each sequence undergoes rigorous quality control through physical feasibility validation and multi-granularity filtering, with detailed second-wise text annotations. We also develop an automated pipeline for creating long-horizon sequences, enabling complex motion generation. Building upon OpenT2M, we introduce no-frill, a pretrained T2M model that achieves excellent performance without complicated designs and technique tricks. Its core component is 2D-PRQ, a novel motion tokenizer that captures spatial and temporal dependencies by dividing the human body into five parts. Comprehensive experiments show that OpenT2M significantly improves generalization of existing T2M models, while 2D-PRQ achieves superior reconstruction and strong zero-shot performance. We expect OpenT2M and no-frill will advance the T2M field by addressing longstanding data quality and benchmarking challenges. Our data and code are released on https://anonymous.4open.science/r/OpenT2M.

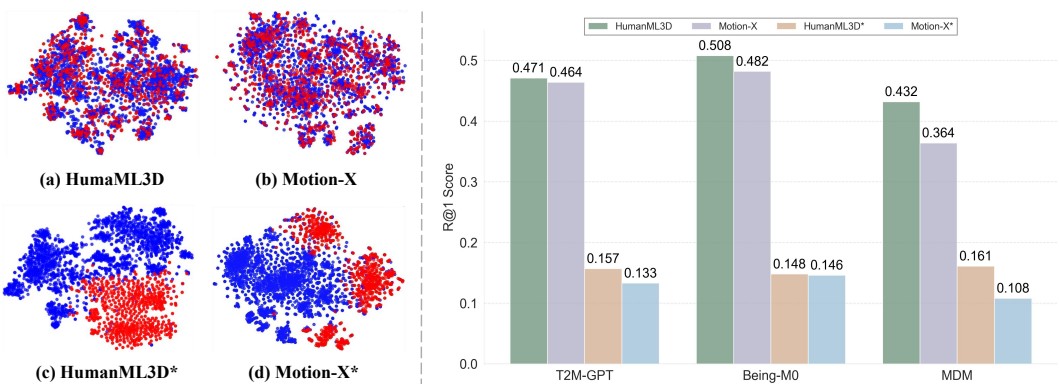

Figure 1: **(left)** Visualization of text description embeddings for the training and validation sets of HumanML3D and Motion-X. A substantial overlap between the sets indicates data leakage. The *∗* denotes our repartitioned versions (e.g., HumanML3D*), where this overlap has been removed. **(right)** Text-to-motion (T2M) performance comparison on the original versus repartitioned benchmarks. The significant performance drop on the cleaned datasets reveals the limited generalization capability of current methods when faced with out-of-domain data.

## 1 INTRODUCTION

Recent years have seen remarkable progress in generating human motion for video games, movies, and humanoid robots. However, current state-of-the-art methods (Guo et al., 2024; Jiang et al., 2023), which depend heavily on motion-capture data (Mahmood et al., 2019), struggle to create

novel motions beyond what they've seen during training. We argue that this limited generalization in text-to-motion (T2M) models comes from fundamental problems with existing datasets: they lack both diversity and scale. In fact, we suppose that many reported improvements on standard benchmarks may simply reflect overfitting to training data rather than real algorithmic advances. To support this claim, we first perform a systematic statistical analysis.

We analyze the text descriptions in two widely-used benchmarks: HumanML3D and Motion-X (Lin et al., 2023). Using CLIP text encoder (Radford et al., 2021) to encode the descriptions, we find significant overlap between training and validation sets (Figure 1). Specifically, 10.62% validation texts in HumanML3D appear word-for-word in the training set and this jumps to 16.97% for Motion-X — most of them correspond to quite similar motions. We also find duplicate descriptions within the validation sets themselves. This data contamination seriously undermines how we evaluate T2M models. To fix this problem, we create a cleaned version of these datasets, called HumanML3D$^*$ and Motion-X$^*$. **As expected, model performance drops significantly on these cleaned benchmarks. Another concerning issue is that modern T2M methods typically need hundreds of training epochs to converge — a sign of overfitting**. Together, these findings suggest that current performance metrics are artificially inflated, leading to models that perform poorly on new tasks.

One straightforward thought is to create larger and more diverse motion datasets. However, progress in high-quality human motion data has stalled since AMASS was released, mainly because professional motion-capture equipment and facilities are extremely expensive. To avoid these costs, recent work (Wang et al., 2024; Fan et al., 2025) has tried extracting human motion from internet videos using existing motion estimation tools (Shin et al., 2024). While web videos provide access to diverse motion patterns, this approach introduces significant noise. Most importantly, a large portion of motions extracted from videos contain physically unrealistic artifacts like foot sliding, body drifting, and limb intersections, which severely limit their usefulness for training reliable motion generation models (Holden, 2024).

To solve these problems, we introduce `OpenT2M`, a large-scale, high-quality human motion dataset containing over one million sequences. Our dataset focuses on bridging the quality gap towards motion-capture databases like HumanML3D while being much larger in scale. The key advantage of `OpenT2M` is that it's freely available to researchers and uses a carefully designed curation process. Unlike previous large-scale video-based motion datasets, which are either not publicly available (Wang et al., 2024) or lack proper physical-aware quality control, we make our dataset open-source with an effective refinement pipeline. To help get started, we're initially releasing 10% of the dataset along with curated text annotations. The download link is provided in the Abstract. `OpenT2M` offers four key improvements over existing datasets. **(1) Physically Feasible Validation:** We validate that all motion sequences are physically feasible and can be simulated, making them suitable for training models that control humanoid robots. **(2) Multi-granularity Quality Filtering:** We remove sequences with occlusions or partial body captures, ensuring that the full human body is visible throughout each motion sequence. **(3) Second-wise Descriptions:** We generate detailed text annotations for every second of motion, then combine them into comprehensive descriptions that accurately capture all actions in the video. **(4) Long-horizon Motions:** Our dataset includes extended motion sequences that enable models to generate realistic, long-term movements from complex text descriptions. In addition, the increasing scale of motion datasets also poses a challenge for motion tokenizers in accurately reconstructing motions. Inspired by residual vector quantization (RQ) techniques (Lee et al., 2022; Guo et al., 2024) and MotionBook (Wang et al., 2024), we propose a novel motion tokenizer, named **2D-PRQ**, that shows superior reconstruction performance and great zero-shot ability. Our contributions are summarized as follows:

- **A Large-scale, High-quality Motion Database.** We curate **`OpenT2M`** containing over one million sequences. Our dataset ensures all motions are physically realistic through multi-granularity quality filtering and manual validation. It also includes long-horizon motion sequences that enable T2M models to generate complex movements from detailed text descriptions.

- **A New Robust Foundation Benchmark.** In addition to improve the generalization of current T2M models, more importantly, `OpenT2M` provides a reliable benchmark for fairly evaluating existing methods.

- **An Effective No-frill T2M Model.** We develop a powerful yet "no-frill" motion generation model that achieves excellent T2M performance without complicated designs or technical trick. Our model, called `no-frill`, uses **2D-PRQ** — a novel motion tokenizer that effectively captures

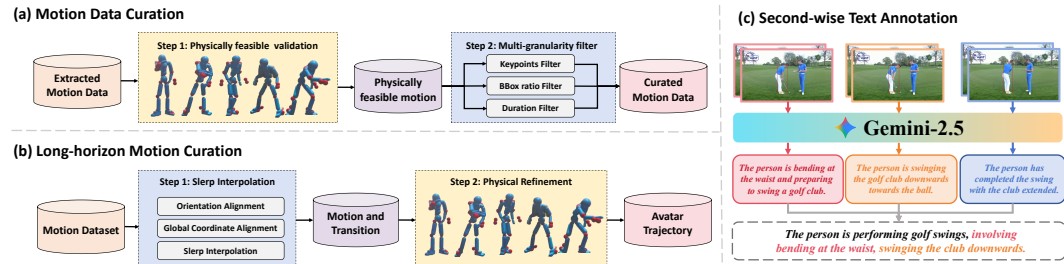

Figure 2: **Data Curation pipeline. (a)** We adopt a two-stage pipeline, including physically feasible validation and multi-granularity filter. **(b)** For text annotation, we generate temporally precise, second-by-second descriptions and synthesize second-wise descriptions into a precise description. **(c)** We adapt the interpolation-based method for motion curation and introduce an RL-policy for refinement.

how motion unfolds both in space and over time. After pretraining on our `OpenT2M`, `no-frill` shows outstanding performance, especially when tested under zero-shot setups.

## 2 RELATED WORK

**Human Motion Dataset.** Dataset is the foundation of building a robust T2M model. Due to pioneering datasets, like KIT (Plappert et al., 2016), AMASS (Mahmood et al., 2019) adapt motion capture devices to obtain human motion data and manual text annotation, the scale and diversity of these datasets are limited. BABEL (Punnakkal et al., 2021) provides frame-level text annotation on AMASS and serves as a long-horizon motion generation benchmark. HumanML3D (Guo et al., 2022) expands human motion datasets with 14.6K motions and 44.9K texts by merging AMASS and HumanAct12 (Guo et al., 2020). Motion-X (Lin et al., 2024) further scales up the dataset by extracting motions from monocular videos and annotating motions by PoseScript (Delmas et al., 2022), resulting in a motion dataset comprising 81.1k sequences. Wang et al. (2024) introduces the first million-level motion dataset, MotionLib, and highlights the importance of scaling datasets. HuMo100M (Cao et al., 2025) is the largest motion dataset featuring 5M motion sequences with multi-granularity text annotation. However, the scarcity of large-scale, high-quality, and open-source datasets hinders building a generalizable T2M model. HumanML3D still contains redundant or highly similar text descriptions. Meanwhile, Motion-X still contains motions that violate physical constraints, such as floating, sliding, and penetration. In this work, we introduce **OpenT2M**, a large-scale, high-quality, and open-source dataset that improves the generalization ability of current T2M models.

**Motion Tokenization.** Building an effective motion tokenizer is crucial for high-quality motion generation. Motion tokenizer contains a motion encoder, a motion decoder, and a quantizer. T2M-GPT (Zhang et al., 2023a) adapts VQ-VAE to discrete motion into motion tokens by applying the 1D convolution and an embedding to represent the whole body feature. Furthermore, to reduce reconstruction error, Lee et al. (2022) introduces residual quantization (RQ), utilizing multiple layers to quantify motion sequences iteratively. Recently, emerging research has explored fine-grained motion tokenization. Chen et al. (2025) decouples the human body into the upper body and lower body, and Cao et al. (2025) decouples the human body into five independent parts. However, these methods encode and quantify different body parts independently without skeletal constraints. This limitation motivates us to design **2D-PRQ**, a novel motion tokenizer capturing spatial and temporal dependencies and showing superior zero-shot performance.

## 3 THE OPENT2M DATASET

The development of robust T2M models is hindered by the lack of large-scale, high-quality data. Prior datasets suffer from insufficient diversity, often leading to the artifact where R@1 exceeds R@1 Real (Zhang et al., 2025; Tanaka et al., 2025; Petrovich et al., 2023), a symptom of ambiguous, one-to-many text-motion mappings. To address this challenge, we introduce **OpenT2M**, an **open-source** dataset created through a rigorous curation pipeline designed with several key steps (Figure 2):

**Physically Feasible Validation.** Motion capture (MoCap) data provides high-quality human motion sequences, valued for its inherent accuracy and adherence to physical constraints (Mahmood et al.,

Table 1: Comparison with existing human motion datasets, where "#physically-feasible" refers to the motion sequences that comply with physical laws and "#long-horizon" denotes the dataset that can serve as a long-horizon benchmark.

| | #Clips | #Hours | #Avg. Length | #long-horizon | #physically-feasible |
|---|---|---|---|---|---|
| BABEL (Punnakkal et al., 2021) | 52.9K | 33.2 | 2.3s | ✓ | ✓ |
| KIT (Plappert et al., 2016) | 5.7K | 11.2 | 9.5s | ✗ | ✓ |
| HumanML3D (Guo et al., 2022) | 29.2K | 28.6 | 7.1s | ✗ | ✗ |
| Motion-X (Lin et al., 2024) | 81.1K | 144.2 | 6.4s | ✗ | ✗ |
| MotionLib (Wang et al., 2024) | 1.2M | 1456.4 | - | ✗ | ✓ |
| MotionMillion (Fan et al., 2025) | 2M | - | - | ✗ | ✗ |
| HuMo100M (Cao et al., 2025) | 5.7M | 8508.3 | 5.3s | ✓ | ✓ |
| OpenT2M | 1M | 2815.6 | **10.1s** | ✓ | ✓ |

2019). However, MoCap data is difficult to scale. To leverage more abundant but noisier video-based motion data, we introduce an RL-based filter to ensure physical plausibility. We train a policy, $\pi_{\text{refine}}$, based on (Luo et al., 2023), to track motions extracted from web videos. By retaining only the motions the policy can successfully track, we eliminate artifacts like jittering and foot-sliding, guaranteeing physical feasibility. Compared with MotionMillion (Fan et al., 2025), we conduct physically feasible validation to ensure extracted motions from web videos adhere to physical constraints, significantly enhancing realism and quality.

**Multi-granularity Filtering.** Web videos are a rich source of human motion (Kay et al., 2017; Wang et al., 2023), but their quality is often compromised by occlusions, blur, and low resolution. To construct a high-fidelity dataset, we extract 2D keypoints using a pre-trained detector (Xu et al., 2022) and apply a set of quality criteria to retain only high-fidelity motions: (1) A minimum keypoint count per frame to ensure structural completeness and reject occluded or partial-body sequences; (2) A minimum bounding box area ratio to guarantee sufficient visibility and detail for accurate motion estimation and text annotation; (3) A minimum motion duration to exclude fragmented clips and retain continuous activities. This pipeline ensures high-quality motion sequences and provides clear video clips for precise text annotation.

**Second-wise Text Annotation.** The precision of text annotations is critical for dataset integrity and motion generation fidelity. Unlike prior works using a single-stage approach (Cao et al., 2025; Wang et al., 2024) to generate a single, coarse description for an entire clip, which fails to capture all activities within the video, leading to the omission of crucial motion details. Our method ensures finer alignment. We implement a two-stage pipeline: first, Gemini-2.5-pro (Team et al., 2024) produces temporal precise, second-by-second descriptions of human motions, including fine-grained limb movements. These fine-grained descriptions are then synthesized into a coherent summary for the entire clip. This process captures comprehensive action details, providing reliable text for building a robust motion generation model.

**Long-horizon Motion Curation.** Existing motion datasets are predominantly short-duration, limiting their utility for long-horizon generation benchmarks. While BABEL (Punnakkal et al., 2021) offers long-horizon motions with fine-grained text labels, its scale and duration remain constrained. To address this, we develop a strategy for synthesizing long-horizon sequences. We first connect raw motions via interpolation with orientation and global coordinate alignment. Since this can create physically implausible transitions, we apply a two-step refinement: an RL-based policy filters out untrackable motions, and we use the avatars' trajectories to ensure physically feasible transitions. In addition, previous work (Cao et al., 2025) creates long-horizon text by directly concatenating annotations, which introduces noise and inefficiency due to motion-irrelevant content. We overcome this by using Gemini-2.5-pro to curate the text: first refining annotations into concise commands, then connecting them to produce clean, user-friendly descriptions. Consequently, `OpenT2M` is the first dataset with an average motion length exceeding 10 seconds. The statistical results of our dataset compared with counterparts are illustrated in Table 1.

## 4 MODEL AND TRAINING

**Overview.** Inspired by large language models' success in multimodal understanding (Luo et al., 2020; Zhang et al., 2024), we frame human motion as a specialized "language". Our approach, illustrated

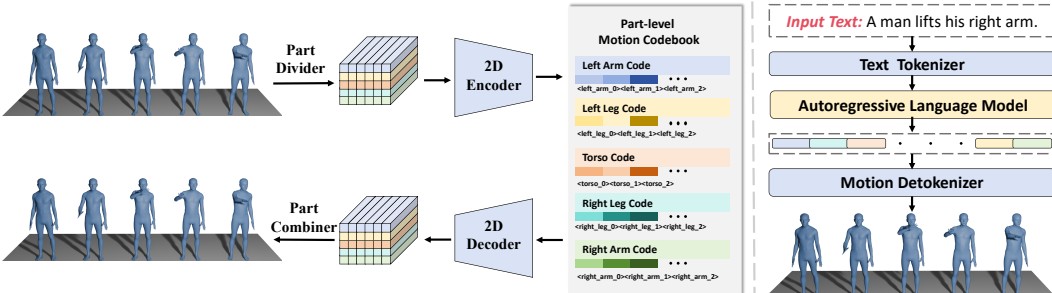

Figure 3: **Model Overview.** We propose an extendable, autoregressive (AR) and discrete T2M model with no frills. **(left)** Our core design 2D-PRQ divides the entire body into five parts, encoding and quantizing motion into a sequence of discrete part-level tokens. **(right)** The AR model takes text as input and predicts part-level motion tokens. We call this model "no-frill" to show its simplicity.

in Figure 3, uses a motion tokenizer to discretize sequences into tokens, which are then generated autoregressively by an LLM. To integrate motion tokens into the LLM backbone, we expand the LLM's vocabulary by incorporating the $K$ discrete codes. We also introduce special tokens such as <mot>, </mot> to delimit motion sequences. The overall training pipeline consists of two phases. First, we train a motion tokenizer to discretize motion features into motion tokens while minimizing reconstruction error. This is followed by a text-motion alignment training via motion instruction tuning (Jiang et al., 2023), which is conducted on OpenT2M to achieve robust and general-purpose text-motion alignment. We name our model as "no-frill" to denote its simplicity and extendable capability without any complex design.

**Motion Instruction Training.** Achieving robust text-motion alignment is essential for developing a generalizable motion generation model. In the text-alignment training phase, 2D-PRQ first encodes and quantizes the continuous raw motion features $\mathcal{M} \in \mathbb{R}^{T \times D}$ sequence of discrete motion tokens $\mathcal{V} \in \mathbb{R}^{n \times p \times l}$, using a temporal downsampling ratio of $n/T$. Here, $p = 5$ represents the number of body parts, $n$ is the number of temporal tokens, and $l$ is the number of residual layers in the quantization process, $K$ is the size of the motion codebook. In addition to common motion tokens, we also introduce another two special tokens <part>, and </part> to separate body-part-specific subsequences in order to structure the input effectively. To enable autoregressive prediction of motion tokens conditioned on descriptions, we design a standardized template for all text-motion pairs:

> **Input** $\mathcal{I}$: The person performs a salute and then shakes hands with another person.
> **Answer** $\mathcal{M}$: <mot> <part_1><motion_token><motion_token> ... < /part_1> ... < /mot>

To train our large motion model, we optimize the negative log-likelihood over the predicted tokens, which is defined as:

$$\mathcal{L}(\Theta) = - \sum_{j=1}^{L} \log P_\Theta(y_j | desc, \hat{y}_{1:j-1}), \tag{1}$$

where $\hat{y}$ and $y$ denote the input and target token sequences, respectively. $\Theta$ represents the model parameters and $L$ is the length of the target sequence.

**2D-PRQ: Towards Generalized Motion Tokenization.** The increasing scale of motion datasets demands more effective encoding. Current VQ-based methods (Zhang et al., 2023b;a) use 1D temporal convolutions and a single embedding for the whole body, leading to information loss and limited generalization. In this work, we propose 2D-PRQ, a novel tokenizer that captures spatiotemporal dependencies by decomposing the body into parts. Given a motion sequence $m_{1:T} \in \mathbb{R}^{T \times D}$, 2D-PRQ first splits it into part-level features $\tilde{m}_{1:T} \in \mathbb{R}^{T \times p \times d}$, where $d$ is the part-level feature dimension, and $p=5$ represents the body parts: {left arm, left leg, torso, right leg, right arm}. Unlike methods that process parts in isolation (Chen et al., 2025), we conceptualize the sequence as a 2D image: time as width and body parts as height. Such design allows us to use a 2D convolution block for motion encoding, capturing both temporal correlations across frames

and spatial dependency between different body parts, which is crucial for maintaining whole-body coordination and consistency. The encoder outputs a latent sequence $\tilde{b}_{1:p;1:n}$ with a downsampling ratio of $n/T$. Each latent vector $\tilde{b}_{i,j}$ is quantized via residual quantization (Lee et al., 2022) using a shared codebook $\mathbb{C}$, producing the token sequence $[b_{1:p;1:n}^k]_{k=0}^K$, where $b^k$ denotes the code sequence at layer $k$. For the decoding, a symmetric 2D decoder reconstructs the part-level features $\hat{m}_{1:p;1:n}$ which are aggregated to restore the raw motion feature $\hat{m}_j$. The reconstruction loss is:

$$\mathcal{L} = \sum^p ||m - \hat{m}||_1 + \sum_{i=0}^p ||m_i - \hat{m}_i||_1 + \beta \sum_{k=1}^K \sum_{i=1}^p ||r_i^k - sg[b_i^j]||_2^2. \tag{2}$$

## 5 EXPERIMENTS

### 5.1 EXPERIMENTAL SETUP

**Datasets.** To evaluate the performance and generalization capabilities of our model, we conduct experiments on three diverse motion datasets: HumanML3D (Guo et al., 2022), Motion-X (Lin et al., 2024), and our collected `OpenT2M`. HumanML3D is a widely adopted benchmark for text-to-motion generation, comprising 4,616 high-quality motion sequences paired with 44,970 textual annotations derived from sources like the AMASS dataset. Motion-X extends this scale with approximately 81,000 motion sequences, incorporating multi-modal data (e.g., video and audio cues) to enhance diversity in complex interactions and long-horizon motions. For further validation on an even larger scale, we utilize `OpenT2M`, a comprehensive dataset with over 1 million motion sequences sourced from real-world human activities, which covers a broad spectrum of human activities, such as walking, dancing, and sports, making it ideal for assessing motion synthesis from diverse language descriptions. Following established protocols, we partition each dataset into training, validation, and test splits using an 80%, 5%, and 15% ratio, respectively.

**Evaluation Metrics.** Our experiments center on two primary tasks to comprehensively assess `no-frill`'s capabilities: text-to-motion (T2M) generation and motion reconstruction. For T2M generation, we adopt standard metrics from the literature (Guo et al., 2022), including Motion-retrieval Precision (R-Precision), Multimodal Distance (MMDist), and Frechet Inception Distance (FID). In addition, the effect of motion tokenizers is assessed by the motion reconstruction task, which reconstructs input motions through the tokenizer to verify discretization quality. We employ FID to measure overall sequence realism and Mean Per Joint Position Error (MPJPE) to quantify geometric accuracy. Details of these metrics can be seen in Appendix B.

**Implementation Details.** For the motion reconstruction task, we implement a motion encoder with a temporal downsampling rate of $\alpha = 4$ for fair comparison. The motion tokenizer is trained with a learning rate of 2e-4 and a batch size of 256. We implement our `no-frill`-2D-PRQ$_4$- with three sizes of LLMs: GPT2-medium (Lagler et al., 2013), LlaMA2-7B (Touvron et al., 2023), and LlaMA3.1-8b (Dubey et al., 2024). Full parameter training is performed on $8 \times$ A800 GPU with a learning rate of 2e-4 and a batch size of 1024 over 5000 steps on `OpenT2M`.

### 5.2 EFFECT OF OPENT2M: GENERALIZATION AND QUICK ADAPTION

While previous works have introduced large-scale datasets (Fan et al., 2025; Wang et al., 2024), their impact on model remains inadequately explored. To address this, we conduct a rigorous T2M evaluation focusing on following key aspects: (1) zero-shot generalization to out-of-domain (OOD) cases, (2) adaptation to novel motion activities via instruction tuning.

**Zero-shot Motion Generalization.** To rigorously assess the generalization of T2M models to unseen data, we curate a held-out evaluation set `OpenT2M`$_{zero}$ comprising 12,000 motions excluded from training data, including HumanML3D and `OpenT2M`, ensuring no domain overlap between the evaluation and training sets. This OOD benchmark enables zero-shot evaluation, where models generate motions for novel text prompts without task-specific fine-tuning. We benchmark three representative baselines: MDM (Tevet et al., 2022), T2M-GPT (Zhang et al., 2023a), Being-M0 (Wang et al., 2024), as well as our `no-frill`. As shown in Table 2, models trained on HumanML3D and Motion-X exhibits limited zero-shot performance, with metrics like FID and R-Precision revealing

degraded semantic alignment and motion diversity on OOD sequences. In contrast, training on `OpenT2M` yields substantial improvements across all baselines, underscoring its role in enhancing generalization through diverse, large-scale coverage of motion primitives and contexts.

Table 2: Comparison of zero-shot performance on $\text{OpenT2M}_{\text{zero}}$ using different datasets for training. Models trained on `OpenT2M` consistently present significant OOD improvements.

| #Model | #training data | R@1 ↑ | R@3 ↑ | FID ↓ | MMDist ↓ | DIV ↑ |
|---|---|---|---|---|---|---|
| Real | - | 0.316 | 0.621 | - | 3.771 | 7.749 |
| MDM | HumanML3D | 0.065 | 0.180 | 51.307 | 7.642 | 3.040 |
| MDM | Motion-X | 0.055 | 0.160 | 56.257 | 8.008 | 3.019 |
| MDM | OpenT2M | **0.194** | **0.447** | **8.153** | **4.889** | **7.136** |
| T2M-GPT | HumanML3D | 0.070 | 0.186 | 62.036 | 8.093 | 2.586 |
| T2M-GPT | Motion-X | 0.063 | 0.173 | 53.464 | 7.770 | 2.957 |
| T2M-GPT | OpenT2M | **0.159** | **0.357** | **5.566** | **5.072** | **6.921** |
| Being-M0 | HumanML3D | 0.073 | 0.190 | 58.541 | 7.956 | 2.932 |
| Being-M0 | Motion-X | 0.057 | 0.157 | 46.222 | 7.652 | 3.220 |
| Being-M0 | OpenT2M | **0.155** | **0.356** | **5.811** | **5.110** | **7.090** |
| no-frill-2D-PRQ$_4$ | HumanML3D | 0.061 | 0.173 | 60.177 | 8.059 | 2.674 |
| no-frill-2D-PRQ$_4$ | Motion-X | 0.052 | 0.152 | 55.47 | 7.841 | 2.433 |
| no-frill-2D-PRQ$_4$ | OpenT2M | **0.240** | **0.512** | **1.475** | **4.281** | **7.563** |

**Motion Instruction Tuning.** Inspired by the two-stage training paradigm in multimodal vision-language models (Liu et al., 2023), we adopt a similar pipeline for T2M generation: an initial pretraining phase on our large-scale `OpenT2M` dataset to foster robust text-motion alignment, followed by targeted fine-tuning on downstream benchmarks. Specifically, we fine-tune the pre-trained model on HumanML3D for a limited 50 epochs. Unlike previous works that train for up to 300 epochs on the same dataset — potentially leading to in-domain overfitting — we intentionally restrict the number of training steps. This allows us to assess inherent generalization capabilities without conflating them with the effects of prolonged training, a potential confound in prior evaluations. As shown in Table 3, models pre-trained on `OpenT2M` consistently outperform their non-pre-trained counterparts, indicating that pre-training equips the model with generalized motion patterns.

Table 3: Comparison of motion instruction tuning on HumanML3D. We apply a limited number of training steps to avoid overfitting. Models with #pretrain consistently achieve significant improvements across diverse #LLM backbones.

| #Model | #LLM backbone | #pretrain | R@1 ↑ | R@3 ↑ | FID ↓ | MMDist ↓ | DIV ↑ |
|---|---|---|---|---|---|---|---|
| Real | - | - | 0.519 | 0.801 | - | 3.176 | 10.954 |
| no-frill | GPT2-medium | - | 0.078 | 0.212 | 61.809 | 8.803 | 4.810 |
| no-frill | LlaMA2-7B | - | 0.472 | 0.741 | 0.619 | 3.572 | 11.226 |
| no-frill | LlaMA3-8B | - | 0.503 | 0.792 | 0.546 | 3.224 | 11.104 |
| no-frill | GPT2-medium | ✓ | 0.215 | 0.377 | 17.91 | 7.129 | 8.372 |
| no-frill | LlaMA2-7B | ✓ | 0.485 | 0.773 | 0.435 | 3.386 | 11.373 |
| no-frill | LlaMA3-8B | ✓ | **0.518** | **0.798** | **0.238** | **3.172** | **11.216** |

## 5.3 EFFECT OF OPENT2M FOR LONG-TERM MOTION GENERATION

Before introducing long-horizon benchmark, we first conduct text refinement. Text annotations in existing datasets, such as HumanML3D, contain considerable redundant details. Directly concatenating texts to construct long-horizon benchmark will introduce noise and inefficiency due to motion-irrelevant content. To mitigate this issue, we

Table 4: Ablation of text refinement on HumanML3D

| #text refinement | R@1 ↑ | R@2 ↑ | R@3 ↑ |
|---|---|---|---|
| - | 0.520 | 0.709 | 0.801 |
| ✓ | **0.533** | **0.720** | **0.808** |

design a specific prompt and utilize Gemini-2.5 to conduct text refinement: (1) removing motion-irrelevant details; (2) converting text annotations into cleaned user commands. As illustrated in Table 4, this text refinement results in an improvement in R-Precision, achieving a better alignment between the refined text and motion sequences.

Following text refinement, we introduce `OpenT2M`$_{long}$, a long-horizon benchmark built with our curation pipeline to evaluate T2M models on extended sequence generation. Our evaluation of a leading model, `no-frill`, reveals a significant struggle to produce satisfied performance without training on long-horizon motion data. In addition, text refinement further substantially improves this ability by enhancing text-motion alignment. Visualizations of the generated sequences are provided in Figure 4, and a detailed comparison with the BABEL dataset is available in Appendix A.2.

Table 5: **Comparison on `OpenT2M`$_{long}$**, where "#text refinement" refers to converting raw texts into cleaned user commands, "#long-horizon" denotes incorporating long-term data into `OpenT2M`.

| Model | #text refinement | #long-horizon | R@1 ↑ | R@3 ↑ | FID ↓ | MMDist ↓ | DIV ↑ |
|---|---|---|---|---|---|---|---|
| Real | - | - | 0.573 | 0.822 | - | 2.842 | 10.450 |
| `no-frill` | - | - | 0.091 | 0.226 | 36.837 | 7.976 | 5.871 |
| `no-frill` | - | ✓ | 0.484 | 0.738 | 0.430 | 3.520 | 10.682 |
| `no-frill` | ✓ | ✓ | **0.510** | **0.765** | **0.297** | **3.322** | **10.748** |

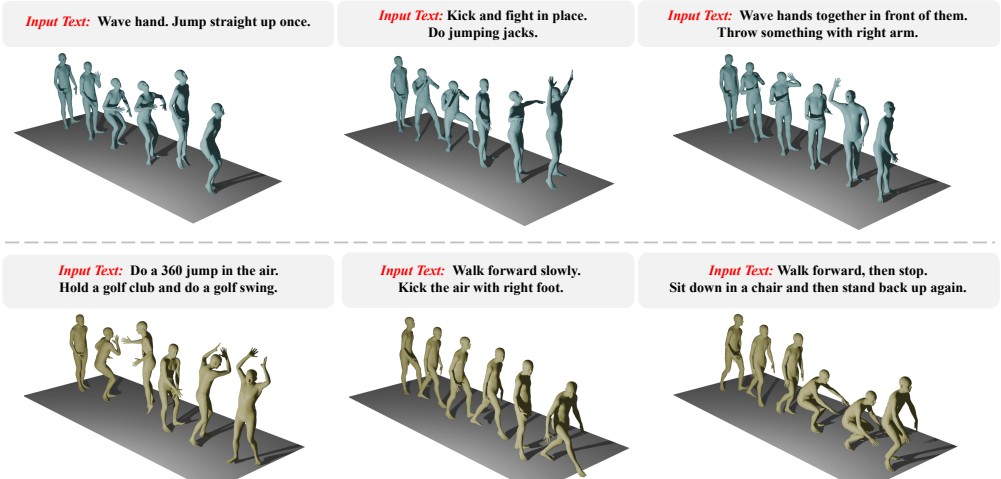

Figure 4: **Visualization of generated long-horizon motions.** Visualization results demonstrate the ability to generate long-horizon motion sequences that accurately align with complex texts.

### 5.4 EFFECT OF MOTION TOKENIZER: 2D-PRQ VS. OTHERS

**Motion Reconstruction Comparison.** As shown in Table 6, our 2D-PRQ tokenizer outperforms previous methods, including PRQ, on large-scale datasets. Under a consistent configuration (codebook size 1024, feature dim 512, except for FSQ (Mentzer et al., 2023)) (codebook size 65536), 2D-PRQ achieves substantially lower reconstruction error on Motion-X and `OpenT2M` while using a simpler architecture. The key advantage lies in its 2D convolutional design, which jointly models spatial and temporal dependencies. This leads to marginal gains on HumanML3D (MPJPE: 25.417 vs. 25.485) but dramatically larger improvements as the dataset scale increases, as evidenced by results on Motion-X (54.493 vs. 73.989) and `OpenT2M` (49.134 vs. 95.743).

**Motion Generation Comparison.** The choice of motion tokenizer is critically dependent on the scale of the training data. As shown in Table 2, replacing VQ-VAE with our 2D-PRQ tokenizer in the Being-M0 model leads to a performance drop when training on smaller datasets like HumanML3D and Motion-X. We attribute this to the increased number of motion tokens in 2D-PRQ, which requires large-scale data for effective training. This hypothesis is confirmed when training on the large-scale

Table 6: Comparison of motion reconstruction. Subscripts denote the number of quantization layers.

| Motion Tokenizer | Codebook Size | HumanML3D FID ↓ | HumanML3D MPJPE ↓ | Motion-X FID ↓ | Motion-X MPJPE ↓ | OpenT2M FID ↓ | OpenT2M MPJPE ↓ |
|---|---|---|---|---|---|---|---|
| $\text{VQ-VAE}_1$ | 1024 | 0.358 | 83.902 | 0.127 | 115.382 | 3.130 | 178.534 |
| $\text{FSQ}_1$ | 65536 | 0.151 | 70.480 | 0.828 | 110.021 | 1.962 | 165.084 |
| $\text{RQ-VAE}_6$ | 1024 | 0.031 | 48.696 | 0.013 | 67.390 | 0.080 | 96.753 |
| $\text{RQ-VAE}_8$ | 1024 | 0.021 | 45.633 | 0.020 | 65.484 | 0.062 | 84.655 |
| $\text{PRQ}_4$ | 1024 | 0.003 | 28.703 | 0.012 | 73.989 | 0.094 | 95.743 |
| $\text{PRQ}_6$ | 1024 | 0.005 | 25.485 | 0.009 | 58.155 | 0.029 | 67.569 |
| $\text{2D-PRQ}_4$ | 1024 | **0.003** | 28.628 | 0.011 | 54.493 | 0.022 | 49.134 |
| $\text{2D-PRQ}_6$ | 1024 | 0.005 | **25.417** | **0.008** | **48.099** | **0.021** | **37.922** |

`OpenT2M`: here, the `no-frill`-$\text{2D-PRQ}_4$ model achieves superior zero-shot performance, even exceeding strong baselines like T2M-GPT, Being-M0, and MDM. This result, also evident in Table 7, underscores that 2D-PRQ unlocks the full potential of large datasets and highlights the critical role of a well-designed motion representation. In Table 7, we observe scaling the LLM from GPT2-medium to LLaMA2-7B brings significant gains. However, further scaling to LLaMA3-8B yields diminishing returns, suggesting a saturation point where performance becomes less dependent on LLM size.

Table 7: Comparison of T2M on `OpenT2M` under different model parameters and motion tokenizers.

| Model | LLM | R@1 ↑ | R@3 ↑ | FID ↓ | MMDist ↓ | DIV ↑ |
|---|---|---|---|---|---|---|
| `no-frill`-$\text{VQ}_1$ | GPT2-medium | 0.257 | 0.513 | 11.226 | 5.146 | 7.393 |
| `no-frill`-$\text{VQ}_1$ | LlaMA2-7B | 0.345 | 0.656 | 3.005 | 3.955 | 8.463 |
| `no-frill`-$\text{VQ}_1$ | LlaMA3-8B | 0.345 | 0.656 | 2.979 | 3.960 | 8.437 |
| `no-frill`-$\text{2D-PRQ}_4$ | GPT2-medium | 0.357 | 0.645 | 8.880 | 4.316 | 7.905 |
| `no-frill`-$\text{2D-PRQ}_4$ | LlaMA2-7B | **0.491** | **0.777** | **0.475** | **2.962** | **9.450** |
| `no-frill`-$\text{2D-PRQ}_4$ | LlaMA3-8B | 0.478 | 0.777 | 0.552 | 3.012 | 8.901 |

**Zero-shot Performance Comparison.** Previous work primarily adopts the $\text{VQ-VAE}_1$ tokenizer and trains it on limited-scale datasets for extensive periods (e.g., 200K steps), which can lead to overfitting and fails to assess the tokenizer's inherent zero-shot generalization ability. In contrast, we pretrain various tokenizers on the large-scale `OpenT2M` dataset and evaluate their zero-shot performance on Hu-

Table 8: Zero-shot comparison of motion tokenizers.

| Motion Tokenizer | HumanML3D FID ↓ | HumanML3D MPJPE ↓ | Motion-X FID ↓ | Motion-X MPJPE ↓ |
|---|---|---|---|---|
| $\text{VQ-VAE}_1$ | 25.525 | 237.702 | 44.889 | 293.301 |
| $\text{PRQ}_4$ | 2.169 | 135.964 | 5.020 | 167.508 |
| $\text{2D-PRQ}_4$ | **0.107** | **77.695** | **1.606** | **108.921** |

manML3D and Motion-X. As shown in Table 8, $\text{2D-PRQ}_4$ significantly outperforms the alternatives, demonstrating its superior generalization and its effectiveness in mitigating tokenizer overfitting.

## 6 CONCLUSION

This paper introduces **OpenT2M**, a large-scale, high-quality human motion dataset with physically feasible validation, multi-granularity filtering, and second-wise annotation. We also introduce a pipeline that synthesizes long-horizon motion autonomously, containing motion connection and text connection to equip T2M models with the capability to generate complex and long-horizon motion sequences. Leveraging `OpenT2M`, we introduce `no-frill`, a pretrained T2M model achieving superior performance without complicated designs. As the core component of `no-frill`, **2D-PRQ**, a novel motion tokenizer, decouples human body features into five parts and captures spatiotemporal dependencies by applying 2D convolution, showing superior reconstruction performance on large-scale datasets and zero-shot ability. Comprehensive experiments demonstrate that `OpenT2M` shows benefits in improving generalization on unseen motion sequences and motion instruction tuning. We hope that our findings and the release of `OpenT2M` will benefit this field.

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

# Appendix

In this appendix, we provide additional details of OpenT2M in Section A. We also provide details of evaluation metrics in Section B. We provide visualization examples of OpenT2M in Section C. Finally, we provide the usage of LLMs in Section D.

## A  DETAILS OF OPENT2M

### A.1  STATISTICAL ANALYSIS OF DATA AND WORD DISTRIBUTION

Figure 5 shows the number distribution of motion sequences across different subsets in OpenT2M on a logarithmic scale, demonstrating variations in dataset sizes. OpenT2M integrates 21 curated subsets, amounting to a comprehensive collection of 1 million motion sequences. A substantial portion of motions in OpenT2M are extracted from web videos utilizing motion estimation models (Shin et al., 2024), such as Kinetics-700 (Kay et al., 2017), Internvid (Wang et al., 2023). These motions undergo rigorous physically feasible validation and multi-granularity filtering. Each motion sequence accounts for over 50% of the duration of the corresponding original video, ensuring temporal consistency and semantic validity. OpenT2M also integrates open-source human motion datasets, such as Motion-X (Lin et al., 2024). Leveraging the proposed long-horizon motion curation pipeline, we construct 190K long-horizon motion sequences. The OpenT2M $_{long}$ comprises motions spliced from two, three, four, and five individual motion sequences. Figure 6 shows the average length distribution of OpenT2M across different subsets. We observe that the dataset with the shortest average sequence length is Postrack, comprising merely 16.12 frames, while 3DPW exhibits the longest average length, exceeding 500 frames. Following a meticulous curation process, OpenT2M exhibits a substantially longer average length compared with previous work (Cao et al., 2025).

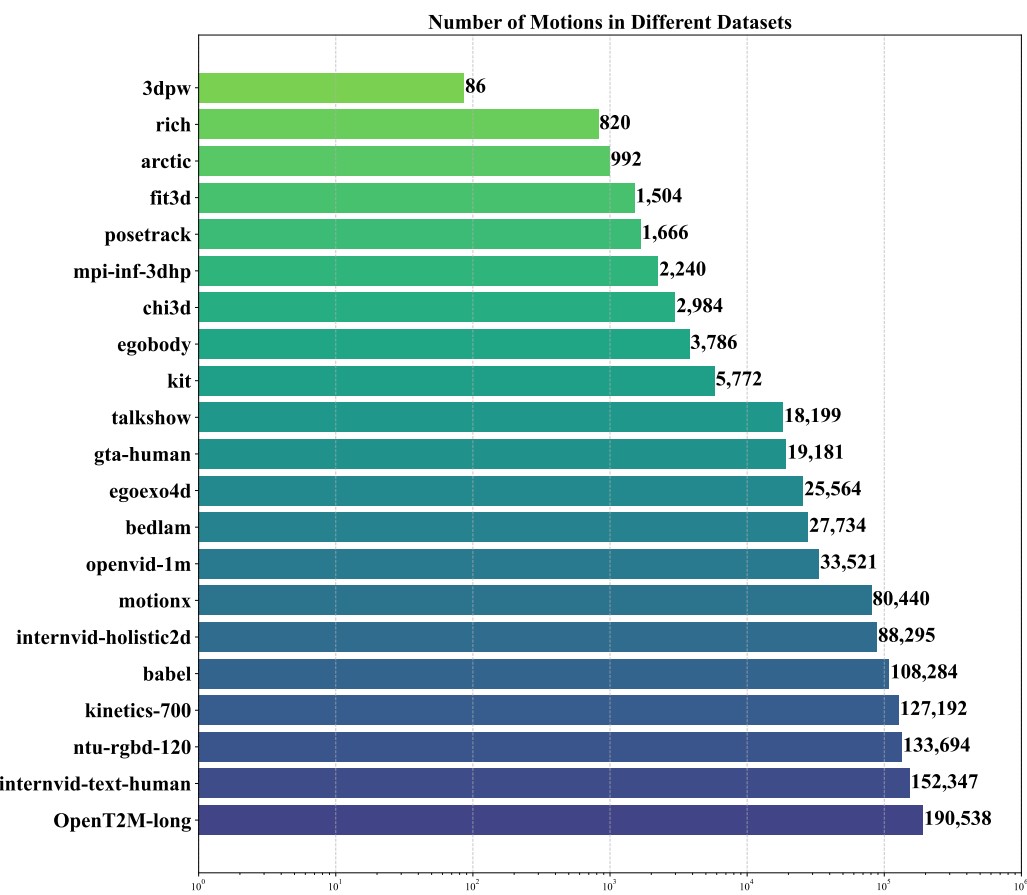

Figure 5: Distribution of motion sequences across different subsets in OpenT2M (logarithmic scale)

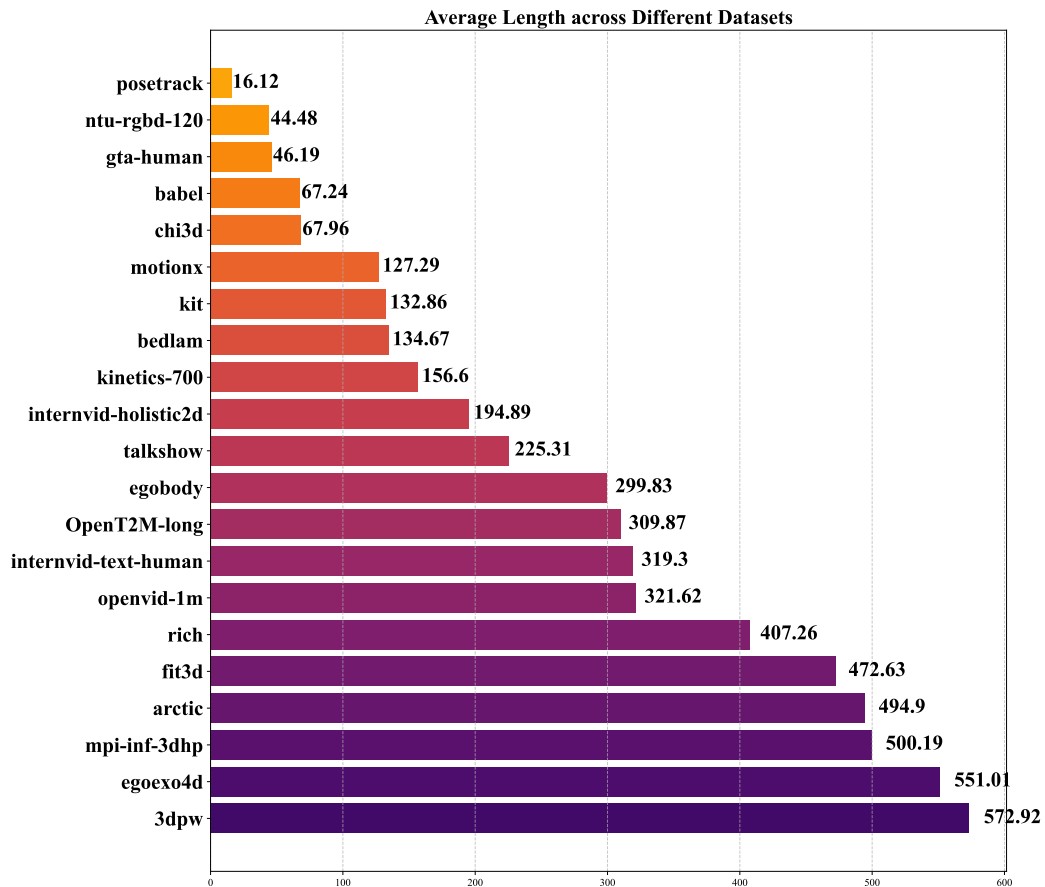

Figure 6: Average Length Distribution of `OpenT2M` across different subsets.

## A.2 LONG-HORIZON MOTION COMPARISON

We first detail the pipeline for long-horizon motion curation. Two different motion sequences are initially aligned in orientation by rotating the initial frame of the second sequence to match the facing direction of the last frame in the first sequence. Subsequently, the entire second sequence is translated spatially to align its position with that of the last frame of the first sequence. Finally, a fixed transition duration is applied, during which spherical linear interpolation is performed between the last frame of the first motion and the initial frame of the second motion to ensure smooth kinematic continuity. To ensure that long-horizon motion sequences adhere to physical constraints, we utilize the concatenated motion sequence as reference poses for an RL policy, driving the avatar in the IsaacGym to track the reference motion. The resulting motion, refined through physical simulation, is adopted as the final long-horizon motion sequences.

Figure 7 shows the length distribution comparison between `OpenT2M`$_{long}$ and BABEL (Punnakkal et al., 2021). BABEL labels about 43 hours of mocap sequences from AMASS (Mahmood et al., 2019) with fine-grained action labels. BABEL exhibits a substantial variation in motion length, containing motion sequences from 5s to over 100s. In BABEL, 37.9% of motion sequences last 5s or less, which significantly limits its effectiveness for evaluating the long-horizon motion generation capability of T2M models. In contrast, `OpenT2M`$_{long}$ contains only 0.33% of motions within 5s. Furthermore, `OpenT2M`$_{long}$ contains 20 times motion sequences than BABEL. As a result, even intervals with relatively low proportions in `OpenT2M`$_{long}$ may contain a larger number of motions compared to BABEL. For instance, motions lasting from 35s to 40s only constitute 0.76% in `OpenT2M`$_{long}$, yet `OpenT2M`$_{long}$ contains 1,454 motion sequences from 35s to 40s. Meanwhile, although the same interval accounts for a higher proportion (0.9%) in BABEL, it represents merely 89 motions.

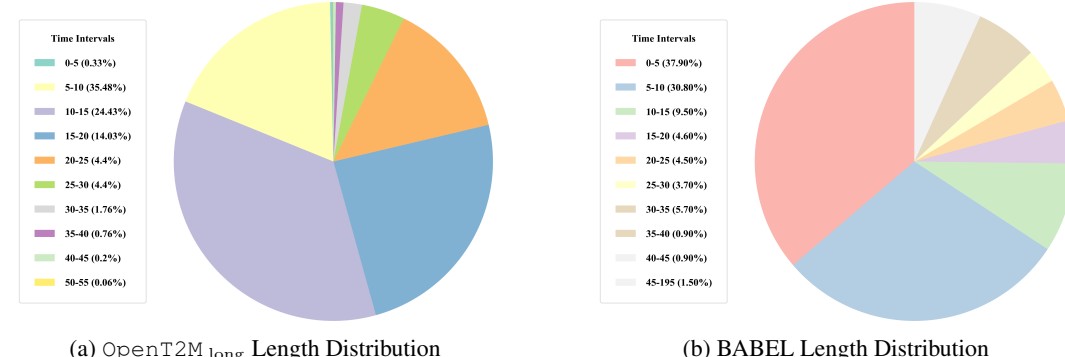

(a) OpenT2M $_{long}$ Length Distribution    (b) BABEL Length Distribution

Figure 7: Length distribution comparison between OpenT2M $_{long}$ and BABEL datasets.

## A.3    SECOND-WISE TEXT ANNOTATION

Previous works (Wang et al., 2024; Cao et al., 2025) typically annotate motion sequences by directly feeding corresponding videos into Vision-Language Models (VLMs) to generate coarse textual descriptions. While this approach offers efficiency, it suffers from a critical limitation: motion sequences extracted from web videos often comprise complex and continuous motion clips. When VLMs are applied in an end-to-end manner to entire video clips, they tend to overlook fine-grained and crucial motion details. Such omissions impact the quality and utility of annotated texts, particularly for applications requiring high temporal precision or detailed kinematic analysis.

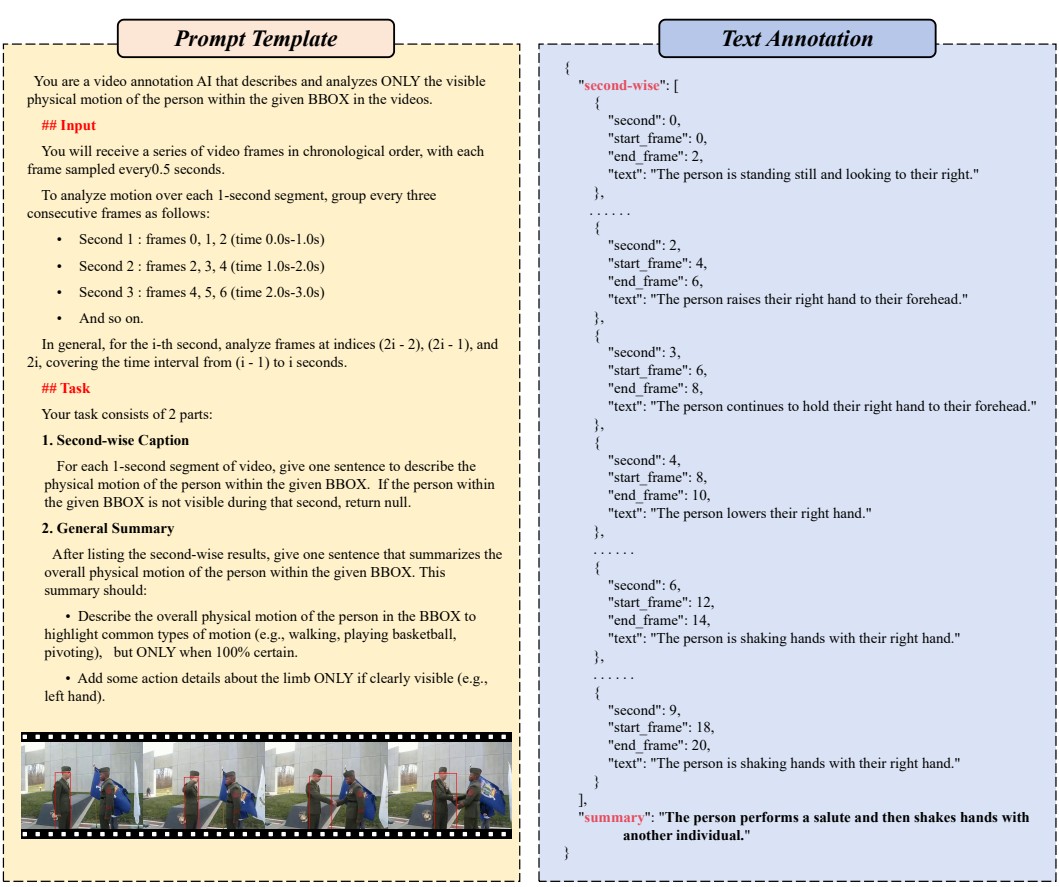

Figure 8: Prompt template for generating second-wise text annotations utilizing Gemini-2.5.

In this work, we design a second-wise annotation scheme as shown in Figure 8. The annotation task mainly contains second-wise captions and a general summary. The process begins by uniformly extracting video frames every 0.5s. Each second video frames are first annotated individually with second-wise descriptions. These second-wise captions are then summarized to form a precise caption for the entire video clip. In the annotation process, we deliberately exclude any descriptions of backgrounds, facial expressions, clothes, and other attributes that are irrelevant to human motion. We computed the word cloud of `OpenT2M`'s text annotations, as shown in Figure 9, revealing that the annotations encompass not only diverse motion patterns but also detailed descriptions of limbs.

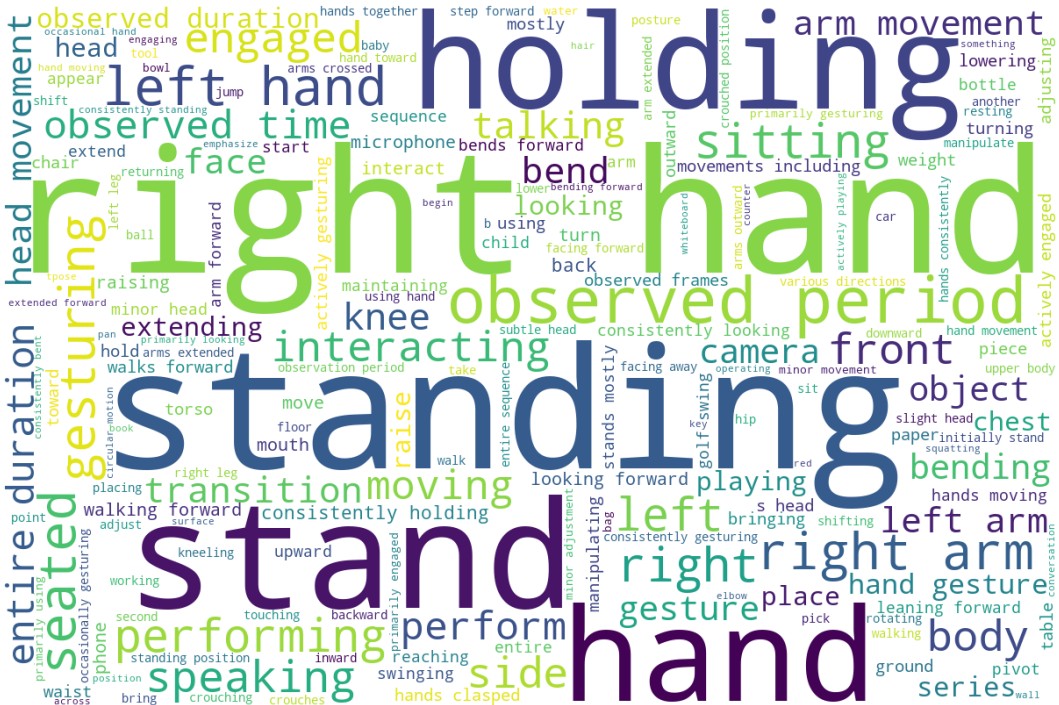

Figure 9: Word cloud visualization of `OpenT2M` text annotations.

## B  EVALUATION METRICS

**Text-to-motion.** We adapt R-precision, MMDist, and FID to evaluate T2M model follow Guo et al. (2022). Each metric is illustrated as follows:

- **R-precision**: The retrieval metric is designed to evaluate the semantic consistency between text and generated motion. The R-precision is computed as the accuracy of its ground-truth text description being ranked Top-1 when retrieved by the generated motion from a text pool. Following Guo et al. (2022), we set the size of the description pool to 32.
- **MMDist:** MultiModel Distance is computed as the average Euclidean distance between motion feature and corresponding text feature.
- **FID**: Frechet Inception Distance is designed to measure the similarity between the distribution of generated motions and ground-truth motion in the feature space. It is computed as the Fréchet distance between the feature distributions of the generated motion and ground-truth motion.

**Motion Reconstruction.** We adapt FID and MPJPE to evaluate motion tokenizers on the motion reconstruction task.

- **FID**: Similar to T2M, Frechet Inception Distance for motion reconstruction is computed as the Fréchet distance between the feature distributions of reconstruction motion and ground-truth motion.

- **MPJPE**: The metric is computed by averaging the L2 distances between all joints of reconstruction motion and ground-truth motion across all frames.

## C  VISUALIZATION EXAMPLES

We provide visualization examples of `OpenT2M` in Figure 10. Visualization examples demonstrate that `OpenT2M` encompasses a diverse range of motion patterns and exhibits strong text-motion alignment, providing a high-quality data foundation for building large motion models.

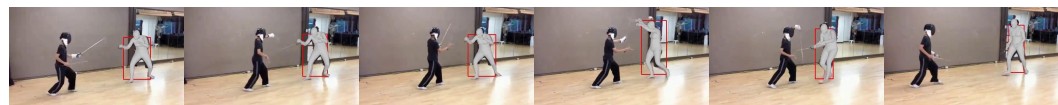

**Text Annotation:** **The person repeatedly lunges forward with their right arm extended and then retracts their arm while stepping back.**

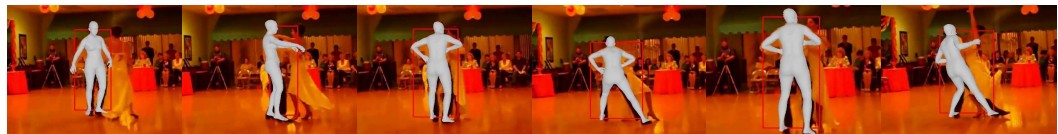

**Text Annotation:** **The person is performing a series of dance moves, involving rotations, leans, and arm extensions.**

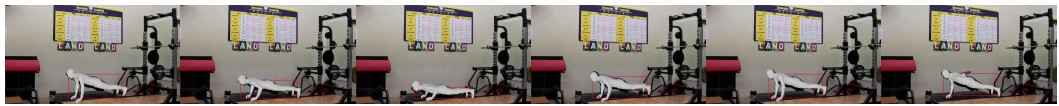

**Text Annotation:** **The person is performing push-ups, moving their chest up and down towards and away from the floor.**

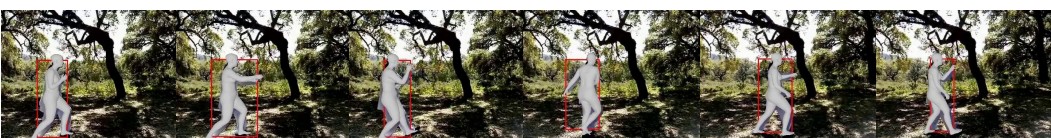

**Text Annotation:** **The person performs a series of slow, deliberate movements, characterized by shifting weight between legs, extending and retracting arms in a flowing motion.**

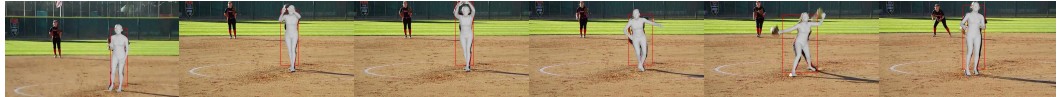

**Text Annotation:** **The person is a softball pitcher who performs a pitching motion including shifting weight, raising their arm, and releasing the ball, followed by recovery.**

Figure 10: Visualization examples of `OpenT2M`, each example is annotated with precise text.

## D  USE OF LARGE LANGUAGE MODELS

In this work, the large language model (LLM) is employed exclusively for text polishing purposes. Its role is limited to refining the linguistic quality, coherence, and stylistic consistency of the textual content, without involvement in data generation and substantive content creation.

