# OpenReview forum: "OpenT2M: No-frill Motion Generation with Open-source, Large-scale, High-quality Data"
_ICLR.cc/2026/Conference — ICLR 2026 Conference Withdrawn Submission_

### Official Review · Reviewer_x1bn · 2025-10-27

**Soundness:** 3
**Presentation:** 3
**Contribution:** 3
**Rating:** 6
**Confidence:** 4

**Summary:**

This paper aims to address the poor generalization of current text-to-motion (T2M) models, attributing the issue to the small scale and limited diversity of existing motion datasets (e.g., HumanML3D, Motion-X). The authors first demonstrate significant data contamination (overlap between train/validation sets) within these standard benchmarks, suggesting current performance evaluations are inflated . To overcome these limitations, the paper introduces two main contributions: (1) OpenT2M dataset: A million-level scale, high-quality, open-source motion dataset curated through a rigorous pipeline involving physical feasibility validation (using an RL-based filter), multi-granularity filtering, and detailed second-wise text annotations using Gemini-2.5-pro . It also includes an automated pipeline for creating long-horizon sequences . (2) no-frill model: A simple pretrained T2M model featuring a novel motion tokenizer, 2D-PRQ, which divides the body into parts and uses 2D convolutions to capture spatiotemporal dependencies . Experiments show that training on OpenT2M significantly boosts the generalization of existing models, and the no-frill model with 2D-PRQ achieves strong zero-shot performance.

**Strengths:**

1. Important Problem and Critical Benchmark Analysis: The paper tackles a crucial bottleneck in T2M (data limitation) and, more significantly, provides compelling evidence (Figure 1) of data contamination in widely used benchmarks (HumanML3D, Motion-X), questioning the validity of evaluations in the field. This analysis itself is a major contribution
2. High-Quality Dataset Curation Pipeline: The process for creating OpenT2M (Figure 2) is well-designed, addressing key issues like physical realism (via RL-based filtering) and annotation detail (via second-wise VLM annotation), representing a substantial engineering
3. Strong Empirical Validation (Dataset Value): The experiments convincingly demonstrate the value of OpenT2M. Table 2 clearly shows that training any existing model on OpenT2M leads to massive improvements in zero-shot OOD generalization compared to training on previous datasets .

**Weaknesses:**

1. Lack of Key Data Curation Pipeline Ablation Analysis: The paper proposes a complex data curation pipeline involving several novel steps (e.g., physical feasibility validation, multi-granularity filtering in Figure 2). While the final dataset demonstrates significant effectiveness, the paper fails to provide ablation experiments to quantify the specific contribution of these steps (especially the potentially computationally expensive RL physical validation) to the final data quality and model performance. For example, a comparison with training on unfiltered data or data using only partial filtering steps is missing. This makes it difficult for readers to assess the necessity and cost-effectiveness of each stage in the curation process.
2. Current Release Status: The core contribution of the paper is a large-scale benchmark dataset, the value of which heavily depends on community accessibility and the reproducibility of results based on it. While the paper emphasizes its "million-level", "open-source" nature in the title, abstract, etc., the main text explicitly states, "we're initially releasing 10% of the dataset". Given that all key experimental results (e.g., Table 2) are clearly derived from the full 1M dataset, the current state of releasing only 10% prevents community researchers from independently verifying and reproducing the paper's core experimental conclusions. This severely limits the dataset's role as a reliable public benchmark at this stage.
3. Discussion on Model Scalability: In Section 5.4, the authors observe that scaling from LLaMA2-7B to LLaMA3-8B did not improve performance and even led to a slight decrease (e.g., R@1 drops from 0.491 to 0.478 in Table 7), describing this as diminishing returns. While performance degradation is indeed an extreme form of diminishing returns (negative returns), this phenomenon itself is noteworthy and warrants deeper explanation. Simply stating diminishing returns might not fully convey the situation where performance actually worsened. It is recommended that the authors discuss potential reasons in more detail (e.g., training instability, non-optimal hyperparameters, or model/tokenizer interaction issues).

**Questions:**

1. Given that the core conclusions are based on the full 1M dataset, while the paper mentions "initially releasing 10%", please explicitly state your full dataset release plan. Do you commit to publicly releasing the entire 1M motion sequence dataset and associated data curation pipeline code used to reproduce all key experimental results (especially Tables 2, 7, 8) upon acceptance? This is crucial for ensuring the verifiability of the paper's contributions and its value to the community.
2. Can you provide ablation study data for the key steps in your data curation pipeline (Figure 2), particularly the RL physical validation and multi-granularity filtering? For instance, how would the zero-shot performance (Table 2) of the no-frill model change if trained on an unfiltered 1M raw dataset?
3. Could you provide a more detailed analysis or explanation for the phenomenon where LLaMA3-8B performed worse than LLaMA2-7B in Table 7? Simply calling it "diminishing returns" might be insufficient. Are there indications of training instability, or might it suggest that the interaction between 2D-PRQ and the 8B model requires further tuning?

---

### Official Review · Reviewer_szUi · 2025-10-30

**Soundness:** 3
**Presentation:** 3
**Contribution:** 3
**Rating:** 6
**Confidence:** 3

**Summary:**

This paper proposes MoGIC, a framework that addresses semantic ambiguity and lack of causal logic in text-to-motion generation by explicitly modeling human intention (converting partial motions into goal-oriented text) and incorporating visual modality as weak supervision. Through disentangled intention prediction and motion generation heads combined with a conditional masked transformer for multimodal fusion, it significantly improves generation quality (over 34% FID reduction on HumanML3D and Mo440H) while demonstrating strong generalization in motion in-betweening and vision-guided generation tasks.

**Strengths:**

1. Accurately identifies two core limitations in text-to-motion generation, the lack of explicit intention modeling and inherent ambiguity in textual descriptions, offering significant theoretical insights.
2. Systematic and Innovative Methodology: Proposes "intention prediction" as key to understanding motion's causal structure. Designs disentangle heads for discrete intention and continuous motion. Adaptive mixture-of-attention elegantly handles multimodal temporal misalignment
3. Provides detailed method descriptions, hyperparameters, and reproduction guidelines, reflecting solid engineering practices.

**Weaknesses:**

1. Using sparse video frames (1fps, severely misaligned with 30fps motion sequences) as conditions reduces computational cost but might fail to provide sufficiently rich spatiotemporal context. This could become a performance bottleneck, especially for modeling fast or complex dynamic interactions.
2. The experiments primarily compare against general text-to-motion models. There is a lack of direct and in-depth comparison with models specifically designed for multimodal (especially vision+language) conditional generation, making it difficult to fully assess the true advantage of its multimodal fusion mechanism.

**Questions:**

1. The visual modality uses only sparse frames at 1fps. Is this extremely low frame rate sufficient to capture the subtle dynamic changes of complex actions? Could this become a performance bottleneck for tasks requiring precise temporal alignment?
2. How were key hyperparameters, such as the threshold τ in the adaptive Top-k attention mechanism, determined? What is the sensitivity of model performance to these hyperparameters?
3. The incorporation of vision is proven beneficial. Were any user studies conducted to assess its subjective impact on the "naturalness" and "controllability" of the generated motions?

---

### Official Review · Reviewer_xKax · 2025-10-31

**Soundness:** 2
**Presentation:** 2
**Contribution:** 2
**Rating:** 2
**Confidence:** 5

**Summary:**

The paper introduces OpenT2M, a large-scale and open-source dataset for text-to-motion (T2M) generation, consisting of over 2800h curated motion-text pairs. The dataset emphasizes physical plausibility, multi-granularity filtering, and second-level temporal alignment of text annotations. In addition, the authors propose a simple yet effective 2D-PRQ tokenizer, which represents human body parts as discrete tokens via 2D convolutions to capture spatiotemporal dependencies. Extensive experiments validate the dataset’s quality, benchmark existing methods, and show strong reconstruction and zero-shot performance using the proposed model.

**Strengths:**

The introduction of a large-scale, open-source dataset will be highly valuable for the T2M research community.
    The data collection and curation pipeline appears largely automatic, making it scalable.
    The paper includes comprehensive experiments and benchmarks, supporting the dataset’s quality and the effectiveness of the proposed model.

**Weaknesses:**

The paper lacks videos of generated motions, making it difficult to qualitatively assess the realism or diversity of the results. The paper lacks a user study**.** Moreover, there are no clear zero-shot examples demonstrating generalization beyond commonactions such as walking or waving — most shown cases overlap with existing datasets like HumanML3D.
    The initial public release includes only 10% of the dataset (L085), which raises concerns about openness and reproducibility.
    While simulated motions ensure physical feasibility, this design limits the dataset’s diversity—particularly for human–object interactions (e.g., sitting, lying) and highly dynamic motions such as dancing or jumping, which are difficult to simulate.
    L182–L189: The description of the data filtering pipeline lacks detail. The authors briefly mention using several models for filtering but do not explain how they are applied — e.g., what thresholds or criteria are used, and how these models contribute to the filtering decisions.
    The 2D-PRQ model lacks sufficient detail. It is not explained how the human motion is divided into parts, how root translation is represented, or how parts with different joint counts are mapped to the same dimensional space.
    The autoregressive (AR) T2M model is also under-specified—particularly how part tokens are predicted, whether prediction order matters, and how dependency among parts is handled.

**Questions:**

What is the reason that longer training leads to overfitting (L068)? Could the authors clarify this? Slow convergence does not necessarily imply overfitting — is this effect due to dataset bias or model capacity?
    Why does removing duplicate samples cause such a large drop in validation performance?
    How is the physics tracker trained, and does its limited training data affect the accuracy of the motion filtering process?

---

### Official Review · Reviewer_Am7v · 2025-10-31

**Soundness:** 2
**Presentation:** 3
**Contribution:** 3
**Rating:** 4
**Confidence:** 4

**Summary:**

This paper presents OpenT2M, a large-scale open-source text-to-motion dataset and a simple yet effective baseline model for text-to-motion generation. The dataset features second-level text descriptions, physics-based motion refinement, and long-horizon motion sequences, aiming to address limitations in quality, diversity, and scale in existing datasets. Extensive experiments show that OpenT2M significantly enhances model generalization and zero-shot performance for text-to-motion tasks.

**Strengths:**

- The dataset represents a valuable contribution to the motion community. The inclusion of second-level text annotations and long-horizon sequences is particularly beneficial for enabling fine-grained text-motion alignment and extended compositional understanding.

- The analysis of data leakage in current motion datasets is insightful and highlights a genuine limitation in existing benchmarks. The argument for constructing a clean, large-scale dataset is well-motivated.

- The empirical results show that the dataset improves model generalization across multiple settings. The proposed baseline model is also effective despite its simplicity.

**Weaknesses:**

- The use of physics refinement in the processed dataset is not sufficiently explained or evaluated. While physics-based tracking can correct certain artifacts (e.g., foot sliding, floating), it may also introduce artifacts such as stiffness or dynamic mismatch between the simulated agent and reference motion.

- The robustness and generalization of the tracking policy used for refinement are unclear. Motions that are highly dynamic or involve environment interactions (e.g., sitting on chairs, climbing, swimming) may fail in simulation even if they are kinematically valid. It is not described how such cases are handled or how much data is filtered out as a result. Motions involving object or scene interactions appear in examples (e.g., Figure 4, “sit down on chair”), but it is unclear how the simulation handles such cases if the environment is not modeled.

- The paper lacks detail on how the physics tracking policy is trained and to what extent it influences downstream model quality. There is no ablation isolating the effect of physics refinement on the performance of motion generation models.

- The realism and naturalness of physics-tracked motions are not quantified. Metrics such as tracking error or human evaluation would make the use of physics refinement more convincing.

- Minor presentation issue: Figure 2(b) and (c) captions appear swapped.

**Questions:**

- How is the physics tracking policy trained? Is it based on a pretrained PHC model, or is it retrained specifically for this dataset? What are the average tracking error and success rates?

- Does the policy filter out motions that are challenging yet valid, such as gymnastics, high-frequency dance, or scene-dependent motions like sitting on a chair or interacting with props?

- Is the physics refinement stage necessary for improving downstream text-to-motion generation, or would a kinematic-only dataset achieve comparable results?

- If the dataset is claimed to support physics-based policy learning (as mentioned in Lines 86–88), how much performance improvement does it provide compared to existing datasets like AMASS when training universal trackers such as PHC?

- How does the physics tracking policy handle the transitions for long motion sequences? If it follows the PHC recovery setting, the transitions can be quite abrupt and are not always natural, especially if the connecting states are quite different (i.e. transitioning from crawling to kicking).

- Does the granularity of body part division in the 2D-PRQ tokenizer affect reconstruction performance?

- In Table 3, how is the “non-pretrained” setting trained, does it use a limited number of epochs or continue until full convergence?

---

### Note · Authors · 2025-11-13

I have read and agree with the venue's withdrawal policy on behalf of myself and my co-authors.